# Indexed Minimum Empirical Divergence for Unimodal Bandits

**Hassan Saber**
Université de Lille, Inria, CNRS, Centrale Lille
UMR 9189 – CRIStAL, F-59000 Lille, France
hassan.saber@inria.fr

**Pierre Ménard**
Otto von Guericke Universität Magdeburg
pierre.menard@ovgu.de

**Odalric-Ambrym Maillard**
Université de Lille, Inria, CNRS, Centrale Lille
UMR 9189 – CRIStAL, F-59000 Lille, France
odalric.maillard@inria.fr

## Abstract

We consider a multi-armed bandit problem specified by a set of one-dimensional family exponential distributions endowed with a unimodal structure. We introduce IMED-UB, an algorithm that optimally exploits the unimodal-structure, by adapting to this setting the Indexed Minimum Empirical Divergence (IMED) algorithm introduced by Honda and Takemura [2015]. Owing to our proof technique, we are able to provide a concise finite-time analysis of the IMED-UB algorithm. Numerical experiments show that IMED-UB competes with the state-of-the-art algorithms.

## 1   Introduction

The multi-armed bandit problem is a popular framework to formalize sequential decision making problems. It was first introduced in the context of medical trials [Thompson, 1933, 1935] and later formalized by Robbins [1952]: A bandit is specified by a configuration, that is a set of unknown probability distributions, $\nu = (\nu_a)_{a \in \mathcal{A}}$ with means $(\mu_a)_{a \in \mathcal{A}}$. At each time $t \in \mathbb{N}$, the learner chooses an arm $a_t \in \mathcal{A}$, based only on the past, the learner then receives and observes a reward $X_t$, conditionally independent, sampled according to $\nu_{a_t}$. The goal of the learner is to maximize the expected sum of rewards received over time (up to some unknown horizon $T$), or equivalently minimize the *regret* with respect to the algorithm constantly receiving the highest mean reward

$$R(\nu, T) = \mathbb{E}_\nu \left[ \sum_{t=1}^{T} \mu^\star - X_t \right] \text{ where } \mu^\star = \max_{a \in \mathcal{A}} \mu_a .$$

Both means and distributions are *unknown*, which makes the problem non trivial, and the learner only knows that $\nu \in \mathcal{D}$ where $\mathcal{D}$ is a given set of bandit configurations. This problem received increased attention in the middle of the 20[th] century, and the seminal paper Lai and Robbins [1985] established the first lower bound on the cumulative regret, showing that designing an algorithm that is optimal uniformly over a given set of configurations $\mathcal{D}$ comes with a price. The study of the lower performance bounds in multi-armed bandits successfully lead to the development of asymptotically optimal algorithms for specific configuration sets, such as the KLUCB algorithm [Lai, 1987, Cappé et al., 2013, Maillard, 2018] for exponential families, or alternatively the DMED and IMED algorithms from Honda and Takemura [2011, 2015]. The lower bounds from Lai and Robbins [1985], later extended by Burnetas and Katehakis [1997] did not cover all possible configurations, and in particular *structured* configuration sets were not handled until Agrawal et al. [1989] and then Graves and Lai [1997] established generic lower bounds. Here, structure refers to the fact that pulling an arm

may reveals information that enables to refine estimation of other arms. Unfortunately, designing numerical efficient algorithms that are provably optimal remains a challenge for many structures.

**Structured configurations.** Motivated by the growing popularity of bandits in a number of industrial and societal application domains, the study of *structured configuration sets* has received increasing attention over the last few years: The linear bandit problem is one typical illustration Abbasi-Yadkori et al. [2011], Srinivas et al. [2010], Durand et al. [2017], for which the linear structure considerably modifies the achievable lower bound, see Lattimore and Szepesvari [2017]. The study of a *unimodal* structure naturally appears in many contexts, e.g. single-peak preference economics, voting theory or wireless communications, and has been first considered in Yu and Mannor [2011] from a bandit perspective, then in Combes and Proutiere [2014] and Trinh et al. [2020] providing an explicit lower bound together with an algorithm exploiting this specific structure. Other structures include Lipschitz bandits Magureanu et al. [2014], and we refer to the manuscript Magureanu [2018] for other examples, such as cascading bandits that are useful in the context of recommender systems. In Combes et al. [2017], a generic algorithm is introduced called `OSSB` (Optimal Structured Stochastic Bandit), stepping the path towards generic multi-armed bandit algorithms that are adaptive to a given structure. More recently in Degenne et al. [2020], the authors introduce an adaptation of the `KLUCB` strategy to handle structured multi-armed bandit problems.

**Unimodal-structure.** We assume a *unimodal* structure similar to that considered in Yu and Mannor [2011] and Combes and Proutiere [2014]. That is, there exists an undirected graph $G = (\mathcal{A}, E)$ whose vertices are arms $\mathcal{A}$, and whose edges $E$ characterize a partial order among means $(\mu_a)_{a \in \mathcal{A}}$. This partial order is assumed unknown to the learner. We assume that there exists a unique optimal arm $a^\star = \mathrm{argmax}_{a \in \mathcal{A}} \mu_a$ and that for all sub-optimal arm $a \neq a^\star$, there exists a path $P_a = (a_1 = a, \ldots, a_{\ell_a} = a^\star) \in \mathcal{A}^{\ell_a}$ of length $\ell_a \geqslant 2$ such that for all $i \in [1, \ell_a - 1]$, $(a_i, a_{i+1}) \in E$ and $\mu_{a_i} < \mu_{a_{i+1}}$. Lastly, we assume that $\nu \subset \mathcal{P} := \{p(\mu), \mu \in \Theta\}$, where $p(\mu)$ is an exponential-family distribution probability with density $f(\cdot, \mu)$ with respect to some positive measure $\lambda$ on $\mathbb{R}$ and mean $\mu \in \Theta \subset \mathbb{R}$. $\mathcal{P}$ is assumed to be known to the learner. Thus, for all $a \in \mathcal{A}$ we have $\nu_a = p(\mu_a)$. We denote by $\mathcal{D}_{(\mathcal{P}, G)}$ or simply $\mathcal{D}$ the structured set of such unimodal-bandit distributions characterized by $(\mathcal{P}, G)$. In the following, we assume that $\mathcal{P}$ is a set of one-dimensional exponential family distributions.

**Contributions.** In this paper, we provide novel regret minimization results related to the unimodal structure. We first revisit the Indexed Minimum Empirical Divergence (IMED) algorithm from Honda and Takemura [2015] introduced for unstructured multi-armed bandits, and adapt it to the unimodal-structured setting. We introduce in Section 3 the `IMED-UB` algorithm that is limited to the pulling of the current best arm or their no more than $d$ nearest arms at each time step, with $d$ the maximum degree of nodes in $G$. Being constructed from IMED, `IMED-UB` does not require any optimization procedure and does not separate exploration from exploitation rounds. `IMED-UB` appears to be a *local* algorithm. We prove in Theorem 6 that `IMED-UB` is asymptotically optimal. Furthermore, this novel algorithm competes with the state-of-the-art algorithms in practice. This is confirmed by numerical illustrations on synthetic data. We believe that the construction of this algorithm together with the proof techniques developed in this paper are of independent interest for the bandit community.

**Notations.** Let $\nu \in \mathcal{D}$. Let $\mu^\star = \max_{a \in \mathcal{A}} \mu_a$ be the optimal mean and $a^\star = \mathrm{argmax}_{a \in \mathcal{A}} \mu_a$ be the optimal arm of $\nu$. We define for an arm $a \in \mathcal{A}$ its sub-optimality gap $\Delta_a = \mu^\star - \mu_a$. Considering an horizon $T \geqslant 1$, thanks to the chain rule we can rewrite the regret as follows:

$$R(\nu, T) = \sum_{a \in \mathcal{A}} \Delta_a \, \mathbb{E}_\nu \big[ N_a(T) \big], \tag{1}$$

where $N_a(t) = \sum_{s=1}^{t} \mathbb{I}_{\{a_s = a\}}$ is the number of pulls of arm $a$ at time $t$.

## 2 Regret lower bound

In this subsection, we recall for completeness the known lower bound on the regret when we assume a unimodal structure. In order to obtain non trivial lower bound we consider algorithms that are *consistent* (aka uniformly-good).

**Definition 1 (Consistent algorithm)** *An algorithm is consistent on $\mathcal{D}$ if for all configuration $\nu \in \mathcal{D}$, for all sub-optimal arm $a$, for all $\alpha > 0$,*

$$\lim_{T \to \infty} \mathbb{E}_\nu \left[ \frac{N_a(T)}{T^\alpha} \right] = 0 \, .$$

We can derive from the notion of consistency an asymptotic lower bound on the regret, see Combes and Proutiere [2014].

**Proposition 2 (Lower bounds on the regret)** *Let us consider a consistent algorithm. Then, for all configuration $\nu \in \mathcal{D}$, it must be that*

$$\liminf_{T \to \infty} \frac{R(\nu, T)}{\log(T)} \geqslant c(\nu) := \sum_{a \in \mathcal{V}_{a^\star}} \frac{\Delta_a}{\mathrm{KL}(\mu_a | \mu^\star)} \, ,$$

*where $\mathrm{KL}(\mu|\mu') = \int_{\mathbb{R}} \log(f(x, \mu)/f(x, \mu')) f(x, \mu) \lambda(\mathrm{d}x)$ denotes the Kullback-Leibler divergence between $\nu = p(\mu)$ and $\nu' = p(\mu')$, for $\mu, \mu' \in \Theta$.*

**Remark 3** *The quantity $c(\nu)$ is a fully explicit function of $\nu$ (it does not require solving any optimization problem) for some set of distributions $\nu$ (see Remark 4). This useful property no longer holds in general for arbitrary structures. Also, it is noticeable that $c(\nu)$ does not involve all the sub-optimal arms but only the ones in $\mathcal{V}_{a^\star}$. This indicates that sub-optimal arms outside $\mathcal{V}_{a^\star}$ are sampled $o(\log(T))$, which contrasts with the unstructured stochastic multi-armed bandits. See Combes and Proutiere [2014] for further insights.*

**Remark 4** *For Bernoulli distributions, a possible setting is to assume $\lambda = \delta_0 + \delta_1$ (with $\delta_0, \delta_1$ Dirac measures), $\Theta = (0, 1)$ and for $\mu \in \Theta$, $f(\cdot, \mu) =: x \in \{0, 1\} \mapsto \mu^x (1 - \mu)^{1-x}$. Then for all $\mu, \mu' \in (0, 1)$, $\mathrm{KL}(\mu|\mu') = \mu \log(\mu/\mu') + (1 - \mu) \log((1 - \mu)/(1 - \mu'))$. For Gaussian distributions (variance $\sigma^2 = 1$), we assume $\lambda$ to be the Lebesgue measure, $\Theta = \mathbb{R}$, and for $\mu \in \mathbb{R}$, $f(\cdot, \mu) =: x \in \mathbb{R} \mapsto (\sqrt{2\pi})^{-1} e^{-(x - \mu)^2/2}$. Then for all $\mu, \mu' \in \mathbb{R}$, $\mathrm{KL}(\mu|\mu') = (\mu' - \mu)^2/2$. For Exponential distributions, we assume $\lambda$ to be the Lebesgue measure, $\Theta = ]0 \, ; +\infty[$, and for $\mu > 0$, $f(\cdot, \mu) =: x > 0 \mapsto e^{-x/\mu}/\mu$. Then for all $\mu, \mu' > 0$, $\mathrm{KL}(\mu|\mu') = \log(\mu'/\mu) + \mu/\mu' - 1$.*

## 3 Optimal algorithm for unimodal-structured bandits

We present in this section a novel algorithm that matches the asymptotic lower bound of Proposition 2. This algorithm is inspired by the Indexed Minimum Empirical Divergence (`IMED`) proposed by Honda and Takemura [2011]. The general idea behind this algorithm is, following the intuition given by the lower bound, to narrow on the current best arm and its neighbourhood for pulling an arm at a given time step.

**Notations.** The empirical mean of the rewards from the arm $a$ is denoted by $\widehat{\mu}_a(t) = \sum_{s=1}^t \mathbb{I}_{\{a_s = a\}} X_s / N_a(t)$ if $N_a(t) > 0$, $0$ otherwise. We also denote by $\widehat{\mu}^\star(t) = \max_{a \in \mathcal{A}} \widehat{\mu}_a(t)$ and $\widehat{\mathcal{A}}^\star(t) = \operatorname*{argmax}_{a \in \mathcal{A}} \widehat{\mu}_a(t)$ respectively the current best mean and the current set of optimal arms.

### 3.1 The `IMED-UB` algorithm.

We first pull each arm once. For all arm $a \in \mathcal{A}$ and time step $t \geqslant 1$ we introduce the `IMED` index

$$I_a(t) = N_a(t) \, \mathrm{KL}(\widehat{\mu}_a(t) | \widehat{\mu}^\star(t)) + \log(N_a(t)) \, ,$$

with the convention $0 \times \infty = 0$. This index can be seen as a transportation cost for moving a sub-optimal arm to an optimal one plus an exploration term: the logarithm of the number of pulls. When an optimal arm is considered, the transportation cost is null and there is only the exploration part. Note that, as stated in Honda and Takemura [2011], $I_a(t)$ is an index in the weaker sense since it cannot be determined only by samples from the arm $a$ but also uses the empirical mean of the current optimal arm. We define `IMED-UB` (Indexed Minimum Empirical Divergence for Unimodal Bandits), described in Algorithm 1, to be the algorithm consisting of pulling an arm $a_t \in \{\widehat{a}_t^\star\} \cup \mathcal{V}_{\widehat{a}_t^\star}$ with

minimum index at each time step $t$, where is $\widehat{a}_t^\star \in \operatorname{argmin}_{\widehat{a}^\star \in \widehat{\mathcal{A}}^\star(t)} N_{\widehat{a}^\star}(t)$ is a current best arm. This is a natural algorithm since the lower bound on the regret given in Proposition 2 involves only the arms in $\mathcal{V}_{a^\star}$, the neighbourhood of the arm $a^\star$ of maximal mean.

---

**Algorithm 1** `IMED-UB`

---

Pull each arm once
**for** $t = |\mathcal{A}| \ldots T - 1$ **do**
    Choose $\widehat{a}_t^\star \in \operatorname*{argmin}_{\widehat{a}^\star \in \widehat{\mathcal{A}}^\star(t)} N_{\widehat{a}^\star}(t)$ (chosen arbitrarily)
    Pull $a_{t+1} \in \operatorname*{argmin}_{a \in \{\widehat{a}_t^\star\} \cup \mathcal{V}_{\widehat{a}_t^\star}} I_a(t)$ (chosen arbitrarily)
**end for**

---

### 3.2 Asymptotic optimality of `IMED-UB`

In this section, we state the main theoretical result of this paper.

**Theorem 5 (Upper bounds)** *Let us consider a set of distributions $\nu \in \mathcal{D}$ and let $a^\star$ its optimal arm. Let $\mathcal{V}_{a^\star}$ be the sub-optimal arms in the neighbourhood of $a^\star$. Then under the `IMED-UB` algorithm for all $0 < \varepsilon < \varepsilon_\nu$, for all horizon time $T \geqslant 1$, for all $a \in \mathcal{V}_{a^\star}$,*

$$\mathbb{E}_\nu[N_a(T)] \leqslant \frac{1 + \alpha_\nu(\varepsilon)}{\mathrm{KL}(\mu_a | \mu_{a^\star})} \log(T) + 2d\, C_\varepsilon \sqrt{\log(c_\varepsilon T)} + d\big(1 + c_{\varepsilon_\nu}^{-1}\big) + d(2d+3)\frac{2\sigma_{\varepsilon_\nu}^2 \, e^{\varepsilon_\nu^2/2\sigma_\varepsilon^2}}{\varepsilon^2} + 1$$

*and, for all $a \notin \{a^\star\} \cup \mathcal{V}_{a^\star}$,*

$$\mathbb{E}_\nu[N_a(T)] \leqslant 2d\, C_\varepsilon \sqrt{\log(c_\varepsilon T)} + d\big(1 + c_{\varepsilon_\nu}^{-1}\big) + d(2d+3)\frac{2\sigma_{\varepsilon_\nu}^2 \, e^{\varepsilon_\nu^2/2\sigma_\varepsilon^2}}{\varepsilon^2} + 1 \,,$$

*where $d$ is the maximum degree of nodes in $G$, $\varepsilon_\nu = \min_{a \neq a'} |\mu_a - \mu_{a'}|/2$, $\sigma_\varepsilon^2 = \max_{a \in \mathcal{A}} \left\{ \mathbb{V}_{X \sim p(\mu')}(X) \colon \mu' \in [\mu_a - \varepsilon, \mu_a] \right\}$ and $c_\varepsilon, C_\varepsilon > 0$ are the constants involved in Theorem 15. $\alpha_\nu(\cdot)$ is a non-negative function depending only on $\nu$ such that $\lim_{\varepsilon \to 0} \alpha_\nu(\varepsilon) = 0$ (see Section 4.1 for more details).*

In particular one can note that the arms in the neighbourhood of the optimal one are pulled $\mathcal{O}(\log(T))$ times while the other sub-optimal arms are pulled $\mathcal{O}\left(\sqrt{\log(T)}\right)$ of times under `IMED-UB`. This is coherent with the lower bound that only involves the neighbourhood of the best arm. More precisely, combining Theorem 5 and the chain rule (1) gives the asymptotic optimality of `IMED-UB` with respect to the lower bound of Proposition 2.

**Corollary 6 (Asymptotic optimality)** *With the same notations as in Theorem 5, then under the `IMED-UB` algorithm*

$$\limsup_{T \to \infty} \frac{R(\nu, T)}{\log(T)} \leqslant c(\nu) = \sum_{a \in \mathcal{V}_{a^\star}} \frac{\Delta_a}{\mathrm{KL}(\mu_a | \mu_{a^\star})} \,.$$

A finite time analysis of `IMED-UB` is provided in following Section 4.

## 4 `IMED-UB` **finite time analysis**

At a high level, the key interesting step of the proof is to realize that the considered algorithm implies empirical lower and empirical upper bounds on the numbers of pulls (see Lemma 7, Lemma 8). Then, based on concentration lemmas (see Section B), the algorithm-based empirical lower bounds ensure the reliability of the estimators of interest (Lemma 12). Interestingly, this makes use of arguments based on recent concentration of measure that enable to control the concentration without adding some $\log \log$ bonus (such a bonus was required for example in the initial analysis of the KL-UCB strategy from Cappé et al. [2013]). Then, combining the reliability of these estimators with the obtained algorithm-base empirical upper bounds, we obtain upper bounds on the average numbers of pulls (Theorem 5). The proof is concise to fit mostly in the next few pages.

### 4.1 Notations

Let us consider $\nu \in \mathcal{D}$ and let us denote by $a^\star$ its best arm. We recall that for all $a \in \mathcal{A}$, $\mathcal{V}_a = \{a' \in \mathcal{A} : (a, a') \in E\}$ is the neighbourhood of arm $a$ in graph $G = (\mathcal{A}, E)$, and that

$$d = \max_{a \in \mathcal{A}} |\mathcal{V}_a|, \quad \varepsilon_\nu = \min_{a \neq a'} \frac{|\mu_a - \mu_{a'}|}{2}. \tag{2}$$

Then, there exists a function $\alpha_\nu(\cdot)$ such that for all $0 < \varepsilon < \varepsilon_\nu$, for all $a \neq a^\star$,

$$\mathrm{KL}(\mu_a + \varepsilon | \mu^\star - \varepsilon) \leqslant (1 + \alpha_\nu(\varepsilon))^{-1} \mathrm{KL}(\mu_a | \mu^\star) \tag{3}$$

and $\lim_{\varepsilon \downarrow 0} \downarrow \alpha_\nu(\varepsilon) = 0$. At each time step $t \geqslant 1$, $\widehat{a}_t^\star$ is arbitrarily chosen in $\operatorname*{argmin}_{a \in \widehat{\mathcal{A}}^\star(t)} N_a(t)$ where $\widehat{\mathcal{A}}^\star(t) = \operatorname*{argmax}_{a \in \mathcal{A}} \widehat{\mu}_a(t)$.

### 4.2 Algorithm-based empirical bounds

The `IMED-UB` algorithm implies inequalities between the indexes that can be rewritten as inequalities on the numbers of pulls. While lower bounds involving $\log(t)$ may be expected in view of the asymptotic regret bounds, we show lower bounds on the numbers of pulls involving instead $\log(N_{a_{t+1}}(t))$, the logarithm of the number of pulls of the current chosen arm. We also provide upper bounds on $N_{a_{t+1}}(t)$ involving $\log(t)$.

We believe that establishing these empirical lower and upper bounds is a key element of our proof technique, that is of independent interest and not *a priori* restricted to the unimodal structure.

**Lemma 7 (Empirical lower bounds)** *Under* `IMED-UB`*, at each step time* $t \geqslant |\mathcal{A}|$*, for all* $a \in \mathcal{V}_{\widehat{a}_t^\star}$*,*

$$\log(N_{a_{t+1}}(t)) \leqslant N_a(t) \, \mathrm{KL}(\widehat{\mu}_a(t) | \widehat{\mu}^\star(t)) + \log(N_a(t)) \tag{4}$$

*and*

$$N_{a_{t+1}}(t) \leqslant N_{\widehat{a}_t^\star}(t). \tag{5}$$

**Proof** For $a \in \mathcal{A}$, by definition, we have $I_a(t) = N_a(t) \mathrm{KL}(\widehat{\mu}_a(t) | \widehat{\mu}^\star(t)) + \log(N_a(t))$, hence

$$\log(N_a(t)) \leqslant I_a(t).$$

This implies, since the arm with minimum index is pulled, $\log(N_{a_{t+1}}(t)) \leqslant I_{a_{t+1}}(t) = \min_{a' \in \{\widehat{a}_t^\star\} \cup \mathcal{V}_{\widehat{a}_t^\star}} I_{a'}(t) \leqslant I_{\widehat{a}_t^\star}(t) = \log(N_{\widehat{a}_t^\star}(t))$. By taking the $\log^{-1}(\cdot)$, the last inequality allows us to conclude. ∎

**Lemma 8 (Empirical upper bounds)** *Under* `IMED-UB` *at each step time* $t \geqslant |\mathcal{A}|$*,*

$$N_{a_{t+1}}(t) \, \mathrm{KL}(\widehat{\mu}_{a_{t+1}}(t) | \widehat{\mu}^\star(t)) \leqslant \log(t). \tag{6}$$

**Proof** As above, by construction we have

$$I_{a_{t+1}}(t) \leqslant I_{\widehat{a}_t^\star}(t).$$

It remains, to conclude, to note that

$$N_{a_{t+1}}(t) \mathrm{KL}(\widehat{\mu}_{a_{t+1}}(t) | \widehat{\mu}^\star(t)) \leqslant I_{a_{t+1}}(t),$$

and

$$I_{\widehat{a}_t^\star}(t) = \log(N_{\widehat{a}_t^\star}(t)) \leqslant \log(t).$$

∎

## 4.3 Non-reliable current means

For all arms $a, a' \in \mathcal{A}$ and for all accuracy $\varepsilon > 0$, let $\mathcal{E}_{a,a'}^+(\varepsilon)$ be the set of times where the current mean of arm $a$ $\varepsilon$-deviates from above while arm $a$ has more pulls than the current pulled arm $a'$,

$$\mathcal{E}_{a,a'}^+(\varepsilon) := \{t \in [\![1, T-1]\!] : a_{t+1} = a', \ N_{a'}(t) \leqslant N_a(t), \ \widehat{\mu}_a(t) \geqslant \mu_a + \varepsilon \}. \tag{7}$$

We similarly define

$$\mathcal{E}_{a,a'}^-(\varepsilon) := \{t \in [\![1, T-1]\!] : a_{t+1} = a', \ N_{a'}(t) \leqslant N_a(t), \ \widehat{\mu}_a(t) \leqslant \mu_a - \varepsilon \}. \tag{8}$$

We also define

$$\mathcal{E}_{a,a'}(\varepsilon) = \mathcal{E}_{a,a'}^+(\varepsilon) \cup \mathcal{E}_{a,a'}^-(\varepsilon). \tag{9}$$

**Definition 9** (KL-log **deviation**) *For $\varepsilon > 0$, the couple of arms $(a, a') \in \mathcal{A}^2$ shows $\varepsilon^-$-KL-log deviation at time step $t \geqslant 1$ if the following conditions are satisfied*

$$\begin{aligned} &(1) \quad a_{t+1} = a' \\ &(2) \quad \widehat{\mu}_a(t) \leqslant \mu_a - \varepsilon \\ &(3) \quad \log(N_{a'}(t)) \leqslant N_a(t) \operatorname{KL}(\widehat{\mu}_a(t)|\mu_a - \varepsilon) + \log(N_a(t)). \end{aligned}$$

For all couple of arms $(a, a') \in \mathcal{A}^2$ and for all accuracy $\varepsilon > 0$, let $\mathcal{K}_{a,a'}^-(\varepsilon)$ be the set of times where couple of arms $(a, a')$ shows $\varepsilon^-$-KL-log deviation, that is

$$\mathcal{K}_{a,a'}^-(\varepsilon) := \left\{ t \in [\![1, T-1]\!] : \begin{array}{ll} (1) & a_{t+1} = a' \\ (2) & \widehat{\mu}_a(t) \leqslant \mu_a - \varepsilon \\ (3) & \log(N_{a'}(t)) \leqslant N_a(t) \operatorname{KL}(\widehat{\mu}_a(t)|\mu_a - \varepsilon) + \log(N_a(t)) \end{array} \right\}. \tag{10}$$

We note that

$$\mathcal{E}_{a,a'}^-(\varepsilon) \subset \mathcal{K}_{a,a'}^-(\varepsilon).$$

We can now resort to concentration arguments in order to control the size of these sets, which yields the following upper bounds. We defer the proof to Appendix A.1.

**Lemma 10 (Bounded subsets of times)** *For $\varepsilon > 0$, for $(a, a') \in \mathcal{A}^2$,*

$$\mathbb{E}_\nu \left[ \left| \mathcal{E}_{a,a'}^+(\varepsilon) \right| \right], \ \mathbb{E}_\nu \left[ \left| \mathcal{E}_{a,a'}^-(\varepsilon) \right| \right] \leqslant \frac{2\sigma_\varepsilon^2 e^{\varepsilon^2/2\sigma_\varepsilon^2}}{\varepsilon^2}$$

$$\mathbb{E}_\nu \left[ \left| \mathcal{K}_{a,a'}^-(\varepsilon) \backslash \mathcal{E}_{a,a'}^-(\varepsilon) \right| \right] \leqslant 1 + c_\varepsilon^{-1} + 2C_\varepsilon \sqrt{\log(c_\varepsilon T)},$$

*where $\sigma_\varepsilon^2 = \max\limits_{a \in \mathcal{A}} \left\{ \mathbb{V}_{X \sim p(\mu')}(X) : \mu' \in [\mu_a - \varepsilon, \mu_a] \right\}$, $c_\varepsilon, C_\varepsilon > 0$ are the constants involved in Theorem 15.*

## 4.4 Non-reliable current best arm

For accuracy $\varepsilon > 0$, let $\mathcal{M}^\star(\varepsilon)$ be the set of times $t \geqslant 1$ that do not belong to $\mathcal{E}_{\widehat{a}_t^\star, a_{t+1}}^+(\varepsilon)$ and where some of the current best arm $\widehat{a}_t^\star$ differs from $a^\star$,

$$\mathcal{M}^\star(\varepsilon) := \left\{ t \geqslant |\mathcal{A}| : \begin{array}{ll} (1) & t \notin \mathcal{E}_{\widehat{a}_t^\star, a_{t+1}}^+(\varepsilon) \\ (2) & \widehat{a}_t^\star \neq a^\star \end{array} \right\}. \tag{11}$$

**Lemma 11 (Relation between subsets of times)** *Under* `IMED-UB`*, for all accuracy $0 < \varepsilon < \varepsilon_\nu = \min\limits_{a \neq a'} |\mu_a - \mu_{a'}|/2$,*

$$\mathcal{M}^\star(\varepsilon) \subset \bigcup_{\substack{t \geqslant 1 \\ a \in \mathcal{V}_{\widehat{a}_t^\star}}} \mathcal{K}_{a, a_{t+1}}^-(\varepsilon_\nu). \tag{12}$$

**Proof** Let us consider $t \in \mathcal{M}^\star(\varepsilon)$. Since $\widehat{a}_t^\star \neq a^\star$, there exists $a \in \mathcal{V}_{\widehat{a}_t^\star}$ such that

$$\mu_a > \mu_{\widehat{a}^\star}. \tag{13}$$

Then, since $\widehat{a}_t^\star \in \operatorname{argmax}_{a \in \mathcal{A}} \widehat{\mu}_a(t)$, we have

$$\widehat{\mu}_{\widehat{a}^\star}(t) = \widehat{\mu}^\star(t) \geqslant \widehat{\mu}_a(t). \tag{14}$$

Since $t \in \mathcal{M}^\star(\varepsilon)$, $t \notin \mathcal{E}_{\widehat{a}_t^\star, a_{t+1}}^+(\varepsilon)$. By considering empirical lower bounds (5) and Equation (7), we have

$$\mu_{\widehat{a}_t^\star} + \varepsilon \geqslant \widehat{\mu}_{\widehat{a}_t^\star}(t). \tag{15}$$

By combining Equations (14) and (15), it comes

$$\mu_{\widehat{a}_t^\star} + \varepsilon \geqslant \widehat{\mu}^\star(t) \geqslant \widehat{\mu}_a(t). \tag{16}$$

Since $\varepsilon < \varepsilon_\nu \leqslant |\mu_a - \mu_{\widehat{a}_t^\star}|/2$, Equation (13) and previous Equation (16) imply

$$\mu_a - \varepsilon_\nu > \widehat{\mu}_{\widehat{a}_t^\star}(t) \geqslant \widehat{\mu}_a(t). \tag{17}$$

Since $a \in \mathcal{V}_{\widehat{a}_t^\star}$, empirical lower bounds (4) imply

$$\log\bigl(N_{a_{t+1}}(t)\bigr) \leqslant N_a(t) \operatorname{KL}(\widehat{\mu}_a(t)|\widehat{\mu}^\star(t)) + \log(N_a(t)). \tag{18}$$

The classical monotonic properties of $\operatorname{KL}(\cdot|\cdot)$ and Equation (17) imply

$$\begin{cases} \widehat{\mu}_a(t) < \mu_a - \varepsilon_\nu \\ \operatorname{KL}(\widehat{\mu}_a(t)|\widehat{\mu}^\star(t)) \leqslant \operatorname{KL}(\widehat{\mu}_a(t)|\mu_a - \varepsilon_\nu). \end{cases} \tag{19}$$

Combining Equations (17) and (19), we get

$$\begin{cases} \widehat{\mu}_a(t) < \mu_a - \varepsilon_\nu \\ \log\bigl(N_{a_{t+1}}(t)\bigr) \leqslant N_a(t) \operatorname{KL}(\widehat{\mu}_a(t)|\mu_a - \varepsilon_\nu) + \log(N_a(t)), \end{cases} \tag{20}$$

which means $t \in \mathcal{K}_{a, a_{t+1}}^-(\varepsilon_\nu)$. ∎

## 4.5 Reliable current means and current best arm

In this subsection, we characterize subsets of times where both the mean of current pulled arm and the optimal mean are well-estimated.

Let us consider for $0 < \varepsilon < \varepsilon_\nu$, for $a \neq a^\star$,

$$\mathcal{U}_a(\varepsilon) = \{t \geqslant |\mathcal{A}| : a_{t+1} = a\} \bigcap \left( \bigcup_{t \geqslant 1} \mathcal{E}_{a_{t+1}, a_{t+1}}^+(\varepsilon) \cup \mathcal{E}_{\widehat{a}_t^\star, a_{t+1}}^-(\varepsilon) \cup \mathcal{E}_{\widehat{a}_t^\star, a_{t+1}}^+(\varepsilon) \cup \mathcal{M}^\star(\varepsilon) \right). \tag{21}$$

Then, Lemma 11 implies

$$\mathcal{U}_a(\varepsilon) \subset \bigcup_{\substack{a' \in \{a\} \cup \mathcal{V}_a \\ a'' \in \mathcal{V}_{a'}}} \mathcal{E}_{a', a}^+(\varepsilon) \cup \mathcal{E}_{a', a}^-(\varepsilon) \cup \mathcal{K}_{a'', a}^-(\varepsilon_\nu). \tag{22}$$

In particular, from Lemma 10 and previous Equation (22) we have

$$\begin{aligned}
\mathbb{E}_\nu[\mathcal{U}_a(\varepsilon)] &\leqslant 2d(d+1) \frac{2\sigma_\varepsilon^2 e^{\varepsilon^2/2\sigma_\varepsilon^2}}{\varepsilon^2} + d\left( \frac{2\sigma_{\varepsilon_\nu}^2 e^{\varepsilon_\nu^2/2\sigma_{\varepsilon_\nu}^2}}{\varepsilon_\nu^2} + 1 + c_{\varepsilon_\nu}^{-1} + 2C_\varepsilon \sqrt{\log(c_\varepsilon T)} \right) \\
&\leqslant d(2d+3) \frac{2\sigma_{\varepsilon_\nu}^2 e^{\varepsilon_\nu^2/2\sigma_\varepsilon^2}}{\varepsilon^2} + d\left( 1 + c_{\varepsilon_\nu}^{-1} + 2C_\varepsilon \sqrt{\log(c_\varepsilon T)} \right), \tag{23}
\end{aligned}$$

where $d = \max_{a \in \mathcal{A}} |\mathcal{V}_a|$ is the maximum degree of nodes in graph $\mathcal{G}$.

**Lemma 12 (Reliable current means)** *Under* IMED-UB, *for all accuracy* $0 < \varepsilon < \varepsilon_\nu = \min_{a \neq a'} |\mu_a - \mu_{a'}|/2$, *for all sub-optimal arm* $a \neq a^\star$, *for all time step* $t \notin \mathcal{U}_a(\varepsilon)$, $t \geqslant |\mathcal{A}|$, *such that* $a_{t+1} = a$,

$$\begin{cases} \widehat{a}_t^\star = a^\star \\ \widehat{\mu}^\star(t) \geqslant \mu^\star - \varepsilon \\ \widehat{\mu}_a(t) \leqslant \mu_a + \varepsilon. \end{cases}$$

## 4.6 Upper bounds on the numbers of pulls of sub-optimal arms

In this subsection, we now combine the different results of the previous subsections to prove Theorem 5.

**Proof** [Proof of Theorem 5.] For $0 < \varepsilon < \varepsilon_\nu$, for $a \neq a^\star$, let us consider $t \notin \mathcal{U}_a(\varepsilon)$, $t \geqslant |\mathcal{A}|$, such that $a_{t+1} = a$. From empirical upper bounds (6), we have

$$N_a(t) \, \mathrm{KL}(\widehat{\mu}_a(t)|\widehat{\mu}^\star(t)) \leqslant \log(t) . \tag{24}$$

From Lemma 12 and Algorithm 1, we have $a \in \mathcal{V}_{a^\star}$ and $\widehat{\mu}_a(t) \leqslant \mu_a + \varepsilon < \mu^\star - \varepsilon \leqslant \widehat{\mu}^\star(t)$. From classical monotonic properties of $\mathrm{KL}(\cdot|\cdot)$ and Equation (3), we have $\mathrm{KL}(\widehat{\mu}_a(t)|\widehat{\mu}^\star(t)) \geqslant \mathrm{KL}(\mu_a + \varepsilon|\mu^\star - \varepsilon) \geqslant (1 + \alpha_\nu(\varepsilon))^{-1} \mathrm{KL}(\mu_a|\mu^\star)$. In view of Equation (24), this implies

$$\forall t \notin \mathcal{U}_a(\varepsilon), t \geqslant |\mathcal{A}| , \text{ such that } a_{t+1} = a, \quad \begin{cases} a \in \mathcal{V}_{a^\star} \\ N_a(t) \leqslant \dfrac{(1 + \alpha_\nu(\varepsilon)) \log(t)}{\mathrm{KL}(\mu_a|\mu^\star)} . \end{cases} \tag{25}$$

For all arm $a \in \mathcal{A}$, for all time step $t \geqslant |\mathcal{A}|$, we denote by

$$\tau_a(t) = \max \{ t' \in [\![ |\mathcal{A}| ; t ]\!] : \; a_{t'+1} = a \quad \text{and} \quad t' \notin \mathcal{U}_a(\varepsilon) \} \tag{26}$$

the last time step before time step $t$ that does not belong to $\mathcal{U}_a(\varepsilon)$ such that we pull arm $a$.

Then, from Equations (25) and (26) we have

$$
\begin{aligned}
\forall a \neq a^\star, \; \forall t \geqslant 1, \quad N_a(t) \; &= \; N_a(|\mathcal{A}|) + \sum_{t' \geqslant |\mathcal{A}|}^{t-1} \mathbb{I}_{\{a_{t'+1} = a\}} \\
&\leqslant \; 1 + \sum_{t' \geqslant 1}^{t-1} \mathbb{I}_{\{a_{t'+1} = a, \, t' \in \mathcal{U}_a(\varepsilon)\}} + \sum_{t' \geqslant |\mathcal{A}|}^{t-1} \mathbb{I}_{\{a_{t'+1} = a, \, t' \notin \mathcal{U}_a(\varepsilon)\}} \\
&\leqslant \; 1 + |\mathcal{U}_a(\varepsilon)| + \sum_{t' \geqslant |\mathcal{A}|}^{t-1} \mathbb{I}_{\{a_{t'+1} = a, \, t' \notin \mathcal{U}_a(\varepsilon)\}} \\
&\leqslant \; 1 + |\mathcal{U}_a(\varepsilon)| + \mathbb{I}_{\{a \notin \mathcal{V}_{a^\star}\}} \times 0 + \mathbb{I}_{\{a \in \mathcal{V}_{a^\star}\}} \times N_a(\tau_a(t)) \\
&\leqslant \; 1 + |\mathcal{U}_a(\varepsilon)| + \mathbb{I}_{\{a \in \mathcal{V}_{a^\star}\}} \frac{(1 + \alpha_\nu(\varepsilon)) \log(\tau_a(t))}{\mathrm{KL}(\mu_a|\mu^\star)} \\
&\leqslant \; 1 + |\mathcal{U}_a(\varepsilon)| + \mathbb{I}_{\{a \in \mathcal{V}_{a^\star}\}} \frac{(1 + \alpha_\nu(\varepsilon)) \log(t)}{\mathrm{KL}(\mu_a|\mu^\star)} .
\end{aligned}
$$

This implies

$$\forall a \neq a^\star, \forall t \geqslant 1, \quad N_a(t) \leqslant \begin{cases} \dfrac{(1 + \alpha_\nu(\varepsilon)) \log(t)}{\mathrm{KL}(\mu_a|\mu^\star)} + |\mathcal{U}_a(\varepsilon)| + 1 & \text{if } a \in \mathcal{V}_{a^\star} \\ |\mathcal{U}_a(\varepsilon)| + 1 & \text{if } a \notin \mathcal{V}_{a^\star} . \end{cases} \tag{27}$$

From Equation (23), averaging these inequalities allows us to conclude. ∎

## 5 Numerical experiments

In this section, we compare empirically the following algorithms : `OSUB`, `UTS` [Combes and Proutiere, 2014, Trinh et al., 2020] and `IMED-UB` described in Algorithm 1. We illustrate how performs the `IMED-UB` algorithm under Bernoulli, Gaussian (variance $\sigma^2 = 0.25$) or Exponential distribution assumption. For the experiments we consider a graph $\mathcal{G}$ with maximal degree $d = 2$ and the unimodal unimodal vectors of means $\mu = (0.05, 0.10, 0.15, 0.20, 0.25, 0.20, 0.15, 0.10, 0.05)$, and average regrets over 500 runs for each distribution family. Based on these experiments (Figure 1), it seems that `IMED-UB` competes with `OSUB` and `UTS`.

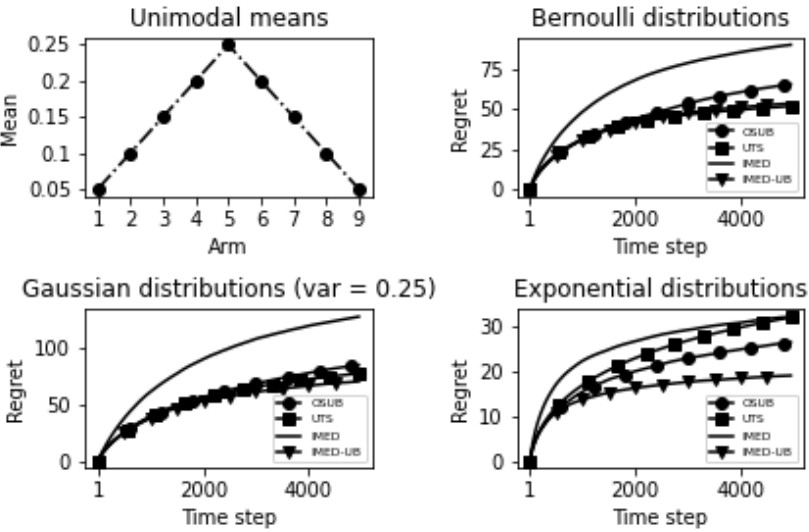

Figure 1: Cumulative regrets averaged over 500 runs.

## Conclusion

In this paper, we have revisited the setup of unimodal multi-armed bandits: We introduced a novel variant based on the `IMED` algorithm. This algorithm does not separate exploration from exploitation rounds and is proven optimal for one-dimensional exponential family distributions. Remarkably, the `IMED-UB` algorithm do not require any optimization procedure, which can be interesting for practitioners. We also provided a novel proof algorithm, in which we make explicit empirical lower and upper bounds, before tackling the handling of bad events by specific concentration tools. This proof technique greatly simplifies and shorten the analysis of `IMED-UB`. Last, we provided numerical experiments that show the practical effectiveness of `IMED-UB`.

## Acknowledgments

This work has been supported by the French Ministry of Higher Education and Research, Inria, the French Agence Nationale de la Recherche (ANR) under grant ANR-16-CE40-0002 (the BADASS project), the MEL, the I-Site ULNE regarding project R-PILOTE-19-004-APPRENF.

Pierre Ménard is supported by the SFI Sachsen-Anhalt for the project REBCI ZS/2019/10/102024 by the Investitionsbank SachsenAnhalt.

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
