## A   IMED-UB **finite time analysis**

We regroup in this section, for completeness, the proofs of the remaining lemmas used in the analysis of IMED-UB in Section 4.

### A.1   Proof of Lemma 10

**Proof** We start by proving $\mathbb{E}_\nu\left[\left|\mathcal{E}_{a,a'}^-(\varepsilon)\right|\right] \leqslant e^{2\varepsilon^2}/2\varepsilon^2$. The proof that $\mathbb{E}_\nu\left[\left|\mathcal{E}_{a,a'}^+(\varepsilon)\right|\right] \leqslant e^{2\varepsilon^2}/2\varepsilon^2$ is similar.

We write

$$\left|\mathcal{E}_{a,a'}^-(\varepsilon)\right| = \sum_{t=1}^{T-1} \mathbb{I}_{\{a_{t+1}=a',\,N_{a'}(t)\leqslant N_a(t),\,\mu_a-\widehat{\mu}_a(t)\geqslant\varepsilon\}} . \tag{28}$$

Considering the stopped stopping times $\tau_n = \inf\{t\geqslant 1, N_{a'}(t)=n\}$ we will rewrite the sum of indicators and use Lemma 14.

$$\begin{aligned}
\left|\mathcal{E}_{a,a'}^-(\varepsilon)\right| &\leqslant \sum_{t\geqslant 1} \mathbb{I}_{\{a_{t+1}=a',\,N_{a'}(t)\leqslant N_a(t),\,\mu_a-\widehat{\mu}_a(t)\geqslant\varepsilon\}} \\
&\leqslant \sum_{n\geqslant 1} \mathbb{I}_{\{n-1\leqslant N_a(\tau_n-1),\,\mu_a-\widehat{\mu}_a(\tau_n-1)\geqslant\varepsilon\}} \\
&\leqslant 1 + \sum_{n\geqslant 2} \mathbb{I}_{\{n-1\leqslant N_a(\tau_n-1),\,\mu_a-\widehat{\mu}_a(\tau_n-1)\geqslant\varepsilon\}} .
\end{aligned} \tag{29}$$

Taking the expectation of Equation (29), it comes

$$\mathbb{E}_\nu\left[\left|\mathcal{E}_{a,a'}^-(\varepsilon)\right|\right] \leqslant 1 + \sum_{n\geqslant 1} \mathrm{P}_\nu\left(\bigcup_{\substack{t\geqslant 1\\N_a(t)\geqslant n}} \widehat{\mu}_a(t) \leqslant \mu_a - \varepsilon\right). \tag{30}$$

From Lemma 14, previous Equation (30) implies

$$\mathbb{E}_\nu\left[\left|\mathcal{E}_{a,a'}^-(\varepsilon)\right|\right] \leqslant 1 + \sum_{n\geqslant 1} \exp(-m\,\mathrm{KL}(\mu_a-\varepsilon|\mu_a)) . \tag{31}$$

From Lemma 13, previous Equation (31) implies

$$\mathbb{E}_\nu\left[\left|\mathcal{E}_{a,a'}^-(\varepsilon)\right|\right] \leqslant \sum_{n\geqslant 0} \exp\left(-n\varepsilon^2/2\sigma_\varepsilon^2\right) = \frac{1}{1-e^{-\varepsilon^2/2\sigma_\varepsilon^2}} , \tag{32}$$

where $\sigma_\varepsilon^2 = \max_{a\in\mathcal{A}}\left\{\mathbb{V}_{X\sim p(\mu')}(X)\colon \mu'\in[\mu_a-\varepsilon,\mu_a]\right\}$. Finally we note that

$$\frac{1}{1-e^{-\varepsilon^2/2\sigma_\varepsilon^2}} = \frac{e^{\varepsilon^2/2\sigma_\varepsilon^2}}{e^{\varepsilon^2/2\sigma_\varepsilon^2}-1} \leqslant \frac{2\sigma_\varepsilon^2 e^{\varepsilon^2/2\sigma_\varepsilon^2}}{\varepsilon^2} .$$

We now show that $\mathbb{E}_\nu\left[\left|\mathcal{K}_{a,a'}^-(\varepsilon)\right|\setminus\left|\mathcal{E}_{a,a'}^-(\varepsilon)\right|\right] \leqslant 1 + c_\varepsilon^{-1} + C_\varepsilon\log\log(c_\varepsilon T)$.

We write

$$\begin{aligned}
&\left|\mathcal{K}_{a,a'}^-(\varepsilon)\setminus\mathcal{E}_{a,a'}^-(\varepsilon)\right| \\
&= \sum_{t=1}^{T-1} \mathbb{I}_{\{a_{t+1}=a',\,1\leqslant N_a(t)<N_{a'}(t),\,\widehat{\mu}_a(t)\leqslant\mu_a-\varepsilon,\,\log(N_{a'}(t))\leqslant N_a(t)\,\mathrm{KL}(\widehat{\mu}_a(t)|\mu_a-\varepsilon)+\log(N_a(t))\}}
\end{aligned} \tag{33}$$

Considering the stopped stopping times $\tau_n = \inf\{t \geqslant 1, N_{a'}(t)=n\}$ we will rewrite the sum $\sum_{t \in [\![1,T-1]\!]} \mathbb{I}_{\{a_{t+1}=a',\, 1 \leqslant N_a(t) < N_{a'}(t),\, \widehat{\mu}_a(t) \leqslant \mu_a - \varepsilon,\, \log(N_{a'}(t)) \leqslant N_a(t)\, \mathrm{KL}(\widehat{\mu}_a(t)|\mu_a - \varepsilon) + \log(N_a(t))\}}$ and use boundary crossing probabilities for one-dimensional exponential family distributions.

$$\left| \mathcal{K}_{a,a'}^-(\varepsilon) \backslash \mathcal{E}_{a,a'}^-(\varepsilon) \right|$$

$$\leqslant \sum_{t=1}^{T-1} \mathbb{I}_{\{a_{t+1}=a',\, 1 \leqslant N_a(t) < N_{a'}(t),\, \widehat{\mu}_a(t) \leqslant \mu_a - \varepsilon,\, \log(N_{a'}(t)) \leqslant N_a(t)\, \mathrm{KL}(\widehat{\mu}_a(t)|\mu_a - \varepsilon) + \log(N_a(t))\}}$$

$$= \sum_{t=1}^{T-1} \sum_{n=1}^{T-1} \mathbb{I}_{\{\tau_{n+1}=t+1\}} \mathbb{I}_{\{1 \leqslant N_a(\tau_{n+1}-1) < n,\, \widehat{\mu}_a(\tau_{n+1}-1) \leqslant \mu_a - \varepsilon\}} \times$$

$$\mathbb{I}_{\{\log(n) \leqslant N_a(\tau_{n+1}-1)\, \mathrm{KL}(\widehat{\mu}_a(\tau_{n+1}-1)|\mu_a - \varepsilon) + \log(N_a(\tau_{n+1}-1))\}}$$

$$= \sum_{n=1}^{T-1} \mathbb{I}_{\{1 \leqslant N_a(\tau_{n+1}-1) < n,\, \widehat{\mu}_a(\tau_{n+1}) \leqslant \mu_a - \varepsilon\}} \times$$

$$\mathbb{I}_{\{\log(n) \leqslant N_a(\tau_{n+1}-1)\, \mathrm{KL}(\widehat{\mu}_a(\tau_{n+1}-1)|\mu_a - \varepsilon) + \log(N_a(\tau_{n+1}-1))\}} \sum_{t=1}^{T-1} \mathbb{I}_{\{\tau_{n+1}=t+1\}}$$

$$\leqslant \sum_{n=1}^{T-1} \mathbb{I}_{\{1 \leqslant N_a(\tau_{n+1}-1) < n,\, \widehat{\mu}_a(\tau_{n+1}) \leqslant \mu_a - \varepsilon,\, \log(n) \leqslant N_a(\tau_{n+1}-1)\, \mathrm{KL}(\widehat{\mu}_a(\tau_{n+1}-1)|\mu_a - \varepsilon) + \log(N_a(\tau_{n+1}-1))\}}$$

$$= \sum_{n=2}^{T-1} \mathbb{I}_{\{1 \leqslant N_a(\tau_{n+1}-1) < n,\, \widehat{\mu}_a(\tau_{n+1}) \leqslant \mu_a - \varepsilon,\, \log(n) \leqslant N_a(\tau_{n+1}-1)\, \mathrm{KL}(\widehat{\mu}_a(\tau_{n+1}-1)|\mu_a - \varepsilon) + \log(N_a(\tau_{n+1}-1))\}} \tag{34}$$

From Equation (34), we get

$$\left| \mathcal{K}_{a,a'}^-(\varepsilon) \backslash \mathcal{E}_{a,a'}^-(\varepsilon) \right| \tag{35}$$

$$\leqslant \sum_{n=2}^{T-1} \mathbb{I}_{\{1 \leqslant N_a(\tau_{n+1}-1) < n,\, \mathrm{KL}(\widehat{\mu}_a(\tau_{n+1}-1)|\mu_a - \varepsilon) \geqslant \log(n/N_a(\tau_{n+1}-1))\}}.$$

Taking the expectation of Equation (35), it comes

$$\mathbb{E}_\nu \left[ \left| \mathcal{K}_{a,a'}^-(\varepsilon) \backslash \mathcal{E}_{a,a'}^-(\varepsilon) \right| \right] \tag{36}$$

$$\leqslant \sum_{n=2}^{T-1} \mathrm{P}_\nu \left( \bigcup_{\substack{t \geqslant 1 \\ \widehat{\mu}_a(t) < \mu_a - \varepsilon \\ 1 \leqslant N_a(t) \leqslant n}} N_a(t)\mathrm{KL}(\widehat{\mu}_a(t)|\mu_a - \varepsilon) \geqslant \log(n/N_a(t)) \right).$$

From Theorem 15, previous Equation (36) implies

$$\mathbb{E}_\nu \left[ \left| \mathcal{K}_{a,a'}^-(\varepsilon) \backslash \mathcal{E}_{a,a'}^-(\varepsilon) \right| \right] \tag{37}$$

$$\leqslant 1 + c_\varepsilon^{-1} + C_\varepsilon \sum_{n \geqslant 1 + c_\varepsilon^{-1}}^{T-1} \frac{c_\varepsilon}{c_\varepsilon n \sqrt{\log(c_\varepsilon n)}}$$

$$\leqslant 1 + c_\varepsilon^{-1} + C_\varepsilon \int_{c_\varepsilon^{-1}}^{T} \frac{c_\varepsilon\, dx}{c_\varepsilon x \sqrt{\log(c_\varepsilon x)}}$$

$$= 1 + c_\varepsilon^{-1} + 2C_\varepsilon \sqrt{\log(c_\varepsilon T)}. \tag{38}$$

∎

## A.2    Proof of Lemma 12

**Proof**  For $0 < \varepsilon < \varepsilon_\nu = \min\limits_{a \neq a'} |\mu_a - \mu_{a'}|/2$, for $a \neq a^\star$, let us consider a time step $t \notin \mathcal{U}_a(\varepsilon)$, $t \geqslant |\mathcal{A}|$ such that $a_{t+1} = a$.

Since $a_{t+1} = a$ and $t \notin \mathcal{U}_{a_{t+1}}(\varepsilon)$ then $t \notin \mathcal{E}^+_{a_{t+1}, a_{t+1}}(\varepsilon)$, that is $\widehat{\mu}_{a_{t+1}}(t) < \mu_{a_{t+1}} + \varepsilon$ or $\widehat{\mu}_a(t) < \mu_a + \varepsilon$ (since $a_{t+1} = a$).

Since $a_{t+1} = a$ and $t \notin \mathcal{U}_{a_{t+1}}(\varepsilon)$ then $t \notin \mathcal{E}^-_{\widehat{a}^\star_t, a_{t+1}}(\varepsilon)$, that is

$$\widehat{\mu}^\star(t) = \widehat{\mu}_{\widehat{a}^\star_t}(t) > \mu_{\widehat{a}^\star_t} - \varepsilon. \tag{39}$$

Since $a_{t+1} = a$ and $t \notin \mathcal{U}_{a_{t+1}}(\varepsilon)$ then $t \notin \mathcal{E}^+_{\widehat{a}^\star_t, a_{t+1}}(\varepsilon) \cup \mathcal{M}^\star(\varepsilon)$. From Equation (11), this implies

$$\widehat{a}^\star_t = a^\star. \tag{40}$$

By combining Equations (39) and (40), we get

$$\widehat{\mu}^\star(t) > \mu_{a^\star} - \varepsilon = \mu^\star - \varepsilon. \tag{41}$$

■

# B    Generic tools

In this section, Pinsker's inequality for one-dimensional exponential family distributions is reminded. Please refer to Lemma 3 from Cappé et al. [2013] for more insights. We also state two concentration results from Maillard [2018]. Relevantly, Theorem 15 is the main concentration result used in this paper.

**Lemma 13 (Pinsker's inequality)**  *For $\mu < \mu'$, it holds that*

$$\mathrm{KL}(\mu|\mu') \geqslant \frac{(\mu' - \mu)^2}{2\sigma^2},$$

*where $\sigma^2 = \max \left\{ \mathbb{V}_{X \sim p(\mu'')}(X) : \mu'' \in [\mu, \mu'] \right\}$.*

**Lemma 14 (Time-uniform concentration)**  *For all arm $a \in \mathcal{A}$, for $x < \mu_a$, $m \geqslant 1$, we have*

$$\mathrm{P}_\nu \left( \bigcup_{\substack{t \geqslant 1 \\ N_a(t) \geqslant m}} \widehat{\mu}_a(t) < x \right) \leqslant \exp(-m\,\mathrm{KL}(x|\mu_a)).$$

**Theorem 15 (Boundary crossing probabilities)**  *For all arm $a \in \mathcal{A}$, for all $\varepsilon > 0$, for all $n \geqslant 1$, we have*

$$\mathrm{P}_\nu \left( \bigcup_{\substack{t \geqslant 1 \\ \widehat{\mu}_a(t) < \mu_a - \varepsilon \\ 1 \leqslant N_a(t) \leqslant n}} N_a(t)\mathrm{KL}(\widehat{\mu}_a(t)|\mu_a - \varepsilon) \geqslant \log(n/N_a(t)) \right) \leqslant \frac{C_\varepsilon}{n\sqrt{\log(c_\varepsilon n)}},$$

*where $c_\varepsilon, C_\varepsilon > 0$ are explained in Maillard [2018].*