# OpenReview forum: "Indexed Minimum Empirical Divergence for Unimodal Bandits"
_NeurIPS.cc/2021/Conference — NeurIPS 2021 Poster_

### Official Review · Reviewer_Sby4 · 2021-07-13

**Rating:** 6
**Confidence:** 3

**Summary:**

The paper tackles the problem of unimodal bandits where it looks at an algorithm based on the IMED index which follows the general idea of the OSUB algorithm. The main novelty of the approach is the analysis which does not rely on distinctions between exploration and exploitation rounds and the added benefit of the IMED indexes not requiring solving an optimisation problem.
The paper is mostly dedicated to the regret analysis but also provides a set of experiments showcasing the numerical performance of the algorithm.

**Limitations And Societal Impact:**

None come to mind.

**Main Review:**

Overall I think this paper falls short due to presentation. My main concern is the clarity of the proof, which contains the main contribution of the work and which I believe is not clearly written and I found hard to assess its correctness (please find suggestions for improvement in the Clarity section).

Originality: The problem solved here has been the target of quite some attention over the past and benefits from a rich literature of results both theoretical and empirical. The originality of the paper comes from applying the IMED index and an analysis that, although follows the same lines as previous regret bound proofs, avoids the need for incorporating dedicated exploration/exploitation rounds in the algorithm. The main architecture of the proof is similar to that of the OSUB algorithm: split all rounds when suboptimal arms are played into sets based on which means are poorly estimated and bound these sets using existing concentration inequalities.

Quality: The paper seems technically sound. The proof feels correct though it is very hard to parse and verify correctness due to how the sets are defined and can probably be substantially clarified. To me it is not obvious whether the sets being bounded in the proof cover all rounds when suboptimal arms are played, though my intuition is that the algorithm should indeed be asymptotically optimal. The experiments show the competitiveness of the algorithm, though not a substantial improvement over the state-of-the-art.

Clarity: Given that the regret analysis constitutes the main contribution of this paper, I think clarity should be substantially improved here. As is the proof leaves a lot of work to the reader to disentangle, in my opinion. I would have liked to see a clear breakdown of how the sets of rounds when suboptimal arms are played being bounded contain all rounds when "bad" arms are played. I suggest this be present in the main architecture of the proof, and I would also recommend starting with the main proof architecture and moving the other supporting lemmas second. There is plenty of space for such a detailed description of how the sets being bounded tie in together. Please also improve the quality of the plots and provide higher resolution graphs - it is hard to read the name of the algorithms as is.

Significance: Overall I feel the results are significant, and could lead to more use of the IMED indexes in bandit algorithms in order to obtain regret guarantees without the extra effort needed to come up with forced exploitation mechanisms.

______________
After rebuttal: I have raised my score to 6 as I feel the results in the paper are significant and the clarity of the paper is not perceived as negatively by the other reviewers (which eases my main concern).

**Time Spent Reviewing:**

6 hours

---

> ### Author Response · Authors · 2021-08-10
> **Reviewer Sby4**
>
> Thank you for the positive words and acknowledging the significance of our work.
>
> Regarding the proof, we are sorry it seems unclear and agree to make a full clean-up of the paper to improve its readability.
>
> Let us highlight that this proof required a significant amount of work, in order to scrutinize every single detail and simplify it weeks after weeks until reaching this fairly short, effective proof: We believe that the fact we can present it entirely in the main body of the paper is indeed a strength of our contribution, when compared with many other theoretical work on multi-armed bandits.
> Also, the resulting proof, although short, contains a few subtle points that we feel are original (and are not present in the proof of OSUB). For instance, the use of the empirical lower bounds substantially differs from what was considered in OSUB.
> Hence, we understand why it is hard to master it only a few reads, given the short amount of time available for a reviewer. Now, we will use the additional page limit to further clarify the proof (in particular explaining why the sets indeed cover all situations), detail more some of its key steps and incorporate your writing suggestions to help the reader.
>
> Regarding the experiments, they are essentially there as a sanity check. Our goal was not to show substantial improvement, but rather that an alternative, algorithmically simpler strategy can indeed be competitve against the state-of-the-art. Also, we believe the proof technique can be useful beyond the unimodal setting and is worth being advertised.

---

### Official Review · Reviewer_8esZ · 2021-07-16

**Rating:** 6
**Confidence:** 3

**Summary:**

The authors propose a novel algorithm for unimodal bandits with stochastic i.i.d. rewards. They prove that this algorithm achieves the information theoretic upper bound derived and attained by the state of the art algorithms. Their algorithm is based on extending the idea of IMED (for classical bandits) to the unimodal settings. They also compare their algorithm with the state of the art using numerical experiments. Overall the results are interesting, the paper is well written and easy to follow, and the analysis seems elegant.

**Main Review:**

Some remarks are below.

1) The authors mention OSSB, which involves computing the regret lower bound, and then attempting to match the Graves-Lai optimization program by computing its solution and then selecting an arm in order to match the constraints of this program. In fact, their algorithm does precisely this, since $I_a(t)$ quantifies the violation of the constraint $N_a(t) KL( \hat \mu_a,\mu^\star) \ge \log(t)$, and the arm with minimal index is selected.

2) In Section 4, the authors argue that the analysis of (Cappé et al,2013) required some $\log \log$ terms to appear for everything to hold. However, as far as I remember, those terms we shown to be necessary, using an argument based on the law of the iterated logarithm. Perhaps the authors could expand on their statement or give a pointer to the place in the paper of (Cappé et al,2013) where the spurious $\log \log$ terms appear.

3) The regret analysis presented in Theorem 5 features additional terms (with respect to the lower bound) and it is not completely clear how large those terms actually might be. More specifically:
- It is understood that the most important regime is when $\epsilon \to 0$. The authors could perhaps explicitly state in the theorem how all the terms such as $\sigma^2_\epsilon$, $\alpha_\epsilon$ when $\epsilon \to 0$ scale in order to assess whether or not they can be neglected in practice. For instance  $\sigma^2_\epsilon \to \mu_a(1-\mu_a)$ in this regime.
- The dependency on the gap  $\epsilon_{nu} = \min_{a,a'} |\mu_a - \mu_{a'}|$ should be highlighted. In fact, when two sub-optimal arms have the same mean, this gap becomes $0$ and the upper bound of Theorem 5 becomes void.
- The bound scales exponentially in $1/Var(X)$, and becomes uninformative whenever some sub-optimal arm has little to no variance. Is this dependency needed, and if so, why ?
- The bound scales as the square degree $d^2$. This $d^2$ term comes from the analysis of Section 4.3, which requires summing over all pairs of arms. Is this dependency necessary ? For instance, if the graph has a degree $d = K/2$ where $K$ is the number of arms, this would imply that the algorithm has a regret scaling like $O(K^2)$ which is  worse than a simple algorithm that ignores the unimodal structure, whose regret would scale as $O(K)$.

4) The numerical experiments seem to be a bit less thorough than the theoretical part, and they do not seem to answer some of the questions left open by the theoretical analysis:
- does the regret actually scale with the square degree ?
- does the regret actually becomes large when two sub-optimal arms have the same mean ?
This would be useful in assessing the link between the theoretical and actual performance. Also, there are no confidence intervals presented.


**Time Spent Reviewing:**

2

---

> ### Author Response · Authors · 2021-08-10
> **Reviewer 8esZ**
>
> 2. The loglog term does not actually relate to the law of iterated logarithm but to a time-uniform concentration bound. Also, the key here is that we can benefit from the positive gap between $\\mu^\\star$ and the sub-optimal means, to allow some inexact concentration (introducing some small $\epsilon$ slack). A second point is that we only need the sum of all resulting error terms to be $o(\\log(T))$. The combination of these two points enables to get rid of the additional loglog term. Such ideas were already present in (Lai, 1988).
>
> 3. We agree to add some paragraph detailing the scaling of the additional terms, to give more intuition. They are all controlled.
> Note that the dependency with $d^2$ and $1/Var(X)$ only appears in a **constant term** (not depending in T). Unlike previous work, we wanted to make explicit all terms (we could have just wrote this is a constant $B(epsilon)$). Now, we believe these scalings are only an artefact of proof, and could perhaps be improved, at the price of much more intricate analysis. Note that this does not affect the asymptotic optimality, and, that here, you are discussing constant additive terms! We already have the first and second order term contolled (which actually scales linearly with d). Having this third order, constant term explicit is actually not trivial, plus does not seem much interesting.
>
> Last, the case when $\epsilon_\nu=0$ is a degenerate case, corresponding to the very peculiar situation when different arms have exact same mean. Our analysis covers the more realistic situation when this is not the case, hence we would need to modify the last part of the analysis to handle the constant additive term differently in such cases. We will add some remark about this specific situation in the camera ready version.
>
>
> 4. We agree to add more experiments, in particular to show that there is no dependency with d^2 and to highlight the performances of the algorithm beyond the case covered by the theory. We will use the additional page for that. Thank you for the suggestion.

---

### Official Review · Reviewer_pkQP · 2021-07-17

**Rating:** 7
**Confidence:** 4

**Summary:**

The paper handles the unimodal bandit setting with reward distributions in an exponential family. The proposed algorithm, IMED-UB, is based on IMED's index and a logarithmic finite-time upper-bound of its regret is proven. The second term in IMED-UB's bound is in $\sqrt{\log T}$, which is worse than the second term in OSUB's and UTS's bounds. However, empirically the proposed algorithm has the same regret or a smaller regret than OSUB and UTS on a linear graph with 9 nodes.

The algorithm and the proof of its regret-bound are interesting. However, some points would benefit from a more thorough investigation.


**Limitations And Societal Impact:**

The potential negative societal impacts were not discussed in the paper, while the bandits are used to fuel recommender systems which, by identifying user's tastes, tend to lock up users in their "bubble".



**Main Review:**

From an algorithmic point of view, the use of IMED's index for the unimodal setting is strait-forward, except for the removal of the exploitation rule. It's worth noting that Equation (5) in Lemma 7 highlights that IMED's index enforces the exploitation of the leader in the same proportions as OSUB's forced exploitation. However, IMED's index adapts the exploitation.

The originality of the paper lies in the proof of its upper-bound which is not a strait-forward combination of OSUB's and IMED's ones. As the corresponding bound is close from the one of state-of-the art algorithms, the paper would benefit from a thorough comparison of these bounds.

Moreover, the authors mention that the proposed algorithm does not require any optimisation procedure. I guess they refer to the optimization procedure required by KL-based index. Again, a more thorough discussion would be much welcome. In the context of unimodal bandits, the index computation is preceded by the identification of the best arm (among $|A|$ arms), while the index is only computed for the $d+1 <= |A|$ arm in the neighborhood of the leader. Which one of both steps is the most costly is application dependent, and I would expect such algorithm to be used in contexts where $d << |A|$.

Finally, the experimental section could include larger and more complex graphs. It could also mention the computation times to support the interest of removing the optimization procedure of KL-based index.

## typos
* L3, L12: aN algorithm
* L31: with algorithms (remove the 'a')
* L91: section A novel algorithm
* L107 where $\hat...$ (remove 'is')
* L113: remove the mention to KLUCB-UB
* Eq (13), Eq (14) : subscript $t$ is missing for $\hat{a}^*$
* L227: 'unimodal' is doubled

# Post-rebuttal and discussions
The reviews and authors' responses strengthened my opinion with respect to this paper.

However, as reviewer Sby4, I consider the presentation of the proof would be enhanced by presenting in the introduction of section 4 the split of iterations:
* $U\_a(\\epsilon)$ vs $[1,T]\\setminus U\_a(\\epsilon)$
* the split of $U\_a(\\epsilon)$ as expressed in Equations (21) and (22)




**Time Spent Reviewing:**

4

---

> ### Author Response · Authors · 2021-08-10
> **Reviewer pkQP**
>
> We thank you for highlighting that the proof of the upper-bound on the regret provided in the paper is not a straightforward combination of OSUB's and IMED's ones.
>
> Equation (5) in Lemma 7 implies in particular that, under IMED-UB, while the current optimal arm, that is the arm with maximal current mean, is not the most pulled arm, there is no exploration: the current best arm is pulled.
>
> The optimisation procedure does indeed refer to the optimization procedure required by KL-based index and IMED-UB algorithm is expected to be used for $d << |\mathcal{A}|$. The identification of the current best arm could be the more time consuming.
>
> Nevertheless, under IMED-UB, explicit and direct calculations with KL function are used to compute the indexes. Under OSUB, for example, KL function must be reversed to compute the indexes. Although the methods to reverse the KL function are well known (dichotomic method ...), they are not generally  explained to the readers in the numerical illustration (the method parameters in particular). We think this is one advantage of the IMED type indexes. We agree to clarify the discussion concerning the optimisation procedure.
>
> We also agree to enrich the experimental section by including experiments with larger and more complex graphs in Appendix that mention the computation times.
>
> The typos will be handled, we thank you.

---

> ### Author Response · Authors · 2021-09-01
> **Reviewer pkQP**
>
> We are pleased that our responses strengthened your opinion with respect to the paper.
>
> Regarding the proof, we agree to improve its readability and we thank you for the proposals for improvements.

---

### Decision · Program_Chairs · 2021-09-27

**Decision:**

Accept (Poster)

**Comment:**

The committee has appreciated the technical value of this paper as well as its significance for the community. We strongly encourage the authors to take into account the remarks that were made on the presentation and writing for the final version of their work.